# ANALYSIS ON GRADIENT PROPAGATION IN BATCH NORMALIZED RESIDUAL NETWORKS

## ABSTRACT

We conduct mathematical analysis on the effect of batch normalization (BN) on gradient backpropogation in residual network training, which is believed to play a critical role in addressing the gradient vanishing/explosion problem, in this work. By analyzing the mean and variance behavior of the input and the gradient in the forward and backward passes through the BN and residual branches, respectively, we show that they work together to confine the gradient variance to a certain range across residual blocks in backpropagation. As a result, the gradient vanishing/explosion problem is avoided. Furthermore, we use the same analysis to discuss the tradeoff between depths and widths of a residual network and demonstrate that shallower yet wider resnets have stronger learning performance that deeper yet thinner resnets.

## 1 INTRODUCTION

Convolutional neural networks (CNNs) (LeCun et al., 1989; Bengio et al., 2009; Krizhevsky et al., 2012) aim at learning a feature hierarchy where higher level features are formed by the composition of lower level features. The deep neural networks act as stacked networks with each layer depending on its previous layer's output. The stochastic gradient descent (SGD) method (Simard et al., 1998) has proved to be an effective way in training deep networks. The training proceeds in steps with SGD, where a mini-batch from a given dataset is fed at each training step. However, one factor that slows down the stochastic-gradient-based learning of neural networks is the internal covariate shift. It is defined as the change in the distribution of network activations due to the change in network parameters during the training.

To improve training efficiency, Ioffe & Szegedy (2015) introduced a batch normalization (BN) procedure to reduce the internal covariate shift. The BN changes the distribution of each input element at each layer. Let $\mathbf{x} = (x_1, x_2, \cdots, x_K)$, be a K-dimensional input to a layer. The BN first normalizes each dimension of $\mathbf{x}$ as

$$x_k^{new} = \frac{x_k - E(x_k)}{\sqrt{Var(x_k)}}, \qquad (1)$$

and then provide the following new input to the layer

$$z_k = \gamma_k x_k^{new} + \beta_k, \qquad (2)$$

where $k = 1, \cdots, K$ and $\gamma_k$ and $\beta_k$ are parameters to be determined. Ioffe & Szegedy (2015) offered a complete analysis on the BN effect along the forward pass. However, there was little discussion on the BN effect on the backpropagated gradient along the backward pass. This was stated as an open research problem in (Ioffe & Szegedy, 2015). Here, to address this problem, we conduct a mathematical analysis on gradient propagation in batch normalized networks.

The number of layers is an important parameter in the neural network design. The training of deep networks has been largely addressed by normalized initialization (Simard et al., 1998; Glo & Bengio, 2015; Saxe et al., 2013; He et al., 2015) and intermediate normalization layers (Ioffe & Szegedy, 2015). These techniques enable networks consisting of tens of layers to converge using the SGD in backpropagation. On the other hand, it is observed that the accuracy of conventional CNNs gets saturated and then degrades rapidly as the network layer increases. Such degradation is not caused by over-fitting since adding more layers to a suitably deep model often results in higher

training errors (Srivastava et al., 2015; He & Sun, 2015). To address this issue, He et al. (2016) introduced the concept of residual branches. A residual network is a stack of residual blocks, where each residual block fits a residual mapping rather than the direct input-output mapping. A similar network, called the highway network, was introduced by Srivastava et al. (2015). Being inspired by the LSTM model (Gers et al., 1999), the highway network has additional gates in the shortcut branches of each block.

There are two major contributions in this work. First, we propose a mathematical model to analyze the BN effect on gradient propogation in the training of residual networks. It is shown that residual networks perform better than conventional neural networks because residual branches and BN help maintain the gradient variation within a range throughout the training process, thus stabilizing gradient-based-learning of the network. They act as a check on the gradients passing through the network during backpropagation so as to avoid gradient vanishing or explosion. Second, we provide insights into wide residual networks based on the same mathematical analysis. The wide residual network was recently introduced by Zagoruyko & Komodakis (2016). As the gradient goes through the residual network, the network may not learn anything useful since there is no mechanism to force the gradient flow to go through residual block weights during the training. In other words, it might be possible that there are only a few blocks that learn useful representations while a large number of blocks share very little information with small contributions to the ultimate goal. We will show that residual blocks that stay dormant are the chains of blocks at the end of each scale of the residual network.

The rest of this paper is organized as follows. Related previous work is reviewed in Sec. 2. Next, we derive a mathematical model for gradient propagation through a layer defined as a combination of batch normalization, convolution layer and ReLU in Sec. 3. Then, we apply this mathematical model to a resnet block in Sec. 4. Afterwards, we use this model to show that the dormant residual blocks are those at the far-end of a scale in deep residual networks in Sec. 5. Concluding remarks and future research directions are given in Sec. 6.

## 2 REVIEW OF RELATED WORK

One major obstacle to the deep neural network training is the vanishing/exploding gradient problem (Bengio et al., 1994). It hampers convergence from the beginning. Furthermore, a proper initialization of a neural network is needed for faster convergence to a good local minimum. Simard et al. (1998) proposed to initialize weights randomly, in such a way that the sigmoid is activated in its linear region. They implemented this choice by stating that the standard deviation of the output of each node should be close to one.

Glo & Bengio (2015) proposed to adopt a properly scaled uniform distribution for initialization. Its derivation was based on the assumption of linear activations used in each layer . Most recently, He et al. (2015) took the ReLU/PReLU activation into consideration in deriving their proposal. The basic principle used by both is that a proper initialization method should avoid reducing or magnifying the magnitude of the input and its gradient exponentially. To achieve this objective, they first initialized weight vectors with zero mean and a certain variance value. Then, they derived the variance of activations at each layer, and equated them to yield an initial value for the variance of weight vectors at each layer. Furthermore, they derived the variance of gradients that are backpropagated at each layer, and equated them to obtain an initial value for the variance of weight vectors at each layer. They either took an average of the two initialized weight variances or simply took one of them as the initial variance of weight vectors. Being built up on this idea, we attempt to analyze the BN effect by comparing the variance of gradients that are backpropagated at each layer below.

## 3 GRADIENT PROPAGATION THROUGH A LAYER

### 3.1 BN LAYER ONLY

We first consider the simplest case where a layer consists of the BN operation only. We use $\mathbf{x}$ and $\tilde{\mathbf{x}}$ to denote a batch of input and output values to and from a batch normalized (BN) layer, respectively. The standard normal variate of $\mathbf{x}$ is $\mathbf{z}$. In gradient backpropagation, the batch of input gradient values to the BN layer is $\Delta\tilde{\mathbf{x}}$ while the batch of output gradient values from the BN layer is $\Delta\mathbf{x}$.

Mathematically, we have

$$\tilde{\mathbf{x}} = BN(\mathbf{x}) \tag{3}$$

By simple manipulation of the formulas given in Ioffe & Szegedy (2015), we can get

$$\Delta x_i = \frac{\gamma}{Std(x_i)}((\Delta\tilde{x}_i - E(\Delta\tilde{x}_i)) - z_i E(\Delta\tilde{x}_i z_i)), \tag{4}$$

where $x_i$ is the $i$th element of batch $\mathbf{x}$ and $Std$ is the standard deviation. Then, it is straightforward to derive

$$E(\Delta x_i) = 0, \quad \text{and} \quad Var(\Delta x_i) = \frac{\gamma^2}{Var(x_i)}(Var(\Delta\tilde{x}_i) - (E(\Delta\tilde{x}_i z_i))^2). \tag{5}$$

## 3.2 CASCADED BN/ReLU/CONV LAYER

Next, we examine a more complex but common case, where a layer consists of three operations in cascade. They are: 1) batch normalization, 2) ReLU activation, and 3) convolution. To simplify the gradient flow calculation, we make some assumptions which will be mentioned whenever needed.

The input to the $L$th Layer of a deep neural network is $\mathbf{y}_{L-1}$ while its output is $\mathbf{y}_L$. We use $BN$, $ReLU$ and $CONV$ to denote the three operations in each sub-layer. Then, we have the following three equations:

$$\tilde{\mathbf{y}}_{L-1} = BN(\mathbf{y}_{L-1}), \quad \hat{\mathbf{y}}_{L-1} = ReLU(\tilde{\mathbf{y}}_{L-1}), \quad \mathbf{y_L} = CONV(\hat{\mathbf{y}}_{L-1}). \tag{6}$$

The relationship between $\mathbf{y}_{L-1}$, $\tilde{\mathbf{y}}_{L-1}$, $\hat{\mathbf{y}}_{L-1}$ and $\mathbf{y}_L$ is shown in Fig. 1. As shown in the figure, $\tilde{\mathbf{y}}_{L-1}$ denotes the batch of output elements from the BN sub-layer. It also serves as the input to the ReLU sub-layer. $\hat{\mathbf{y}}_{L-1}$ denotes the batch of output elements from the ReLU sub-layer. It is fed into the convolution sub-layer. Finally, $\mathbf{y}_L$ is the batch of output elements from the CONV sub-layer. Gradient vectors have $\Delta$ as the prefix to their corresponding vectors in the forward pass. In this figure, $\mathbf{W}_L^f$ is the weight vector of the convolution layer seen by the input and $\mathbf{W}_L^b$ is the weight vector of the convolution layer seen by the back-propagating gradient. The dimensions of $\mathbf{y}_L$ and $\Delta\mathbf{y}_L$ are $n_L$ and $n'_L$, respectively. $y_{L-1,i}$ denotes the $i$th feature of activation $\mathbf{y}_{L-1}$. The variance and mean of any activation or gradient is always calculated across a batch of activations or gradients because we adopt the batch as a representative of the entire sample.

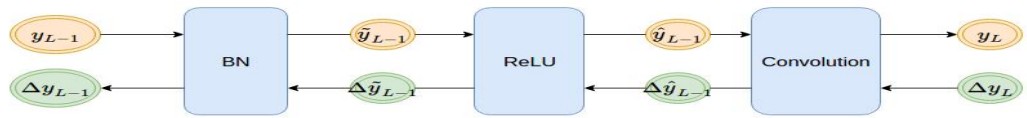

Figure 1: Illustration of a layer that consists of BN, ReLU and CONV three sub-layers.

## 3.3 VARIANCE ANALYSIS IN FORWARD PASS

We will derive the mean and variance of output $y_{L,i}$ from the input $\mathbf{y}_{L-1}$. First, we examine the effect of the BN sub-layer. The output of a batch normalization layer is $\gamma_i z_i + \beta_i$, where $z_i$ is the standard normal variate of $y_{L-1,i}$, calculated across a batch of activations. Clearly, we have

$$E(\tilde{y}_{L-1,i}) = \beta_i, \quad \text{and} \quad Var(\tilde{y}_{L-1,i}) = \gamma_i^2. \tag{7}$$

Next, we consider the effect of the ReLU sub-layer. Let $a = \frac{\beta_i}{\gamma_i}$. We assume that $a$ is small enough so that the standard normal variate $z_i$ follows a nearly uniform distribution in interval $(0, a)$. In Appendix A, we show a step-by-step procedure to derive the mean and variance of the output of the ReLU sub-layer when it is applied to the output of a BN layer. Here, we summarize the main results

below:

$$E(\hat{y}_{L-1,i}) = \gamma_i \left( \frac{1}{\sqrt{2\pi}} + \frac{a}{2} + \frac{a^2}{2} \frac{1}{\sqrt{2\pi}} \right) \tag{8}$$

$$E(\hat{y}_{L-1,i}^2) = \gamma_i^2 \left( 0.5 + \sqrt{\frac{2}{\pi}} a + 0.5a^2 + \frac{a^3}{3\sqrt{2\pi}} \right). \tag{9}$$

Finally, we consider the influence of the CONV sub-layer. To simplify the analysis, we assume that all elements in $\mathbf{W}_L^f$ are mutually independent and with the same distribution of mean 0 and all elements in $\mathbf{y}_{L-1}$ are also mutually independent and with the same distribution across a batch of activations. Furthermore, $\mathbf{y}_{L-1}$ and $\mathbf{W}_L^f$ are independent of each other. Then, we get

$$Var(y_{L,i}) = n_L Var(W_{L,i}^f) E((\hat{y}_{L-1,i})^2). \tag{10}$$

Note that assuming the weight elements come from a distribution with mean 0 is a fair assumption because we initialize the weight elements from a distribution with mean 0 and in the next section, we see that the mean of gradient that reaches the convolution layer during backpropagation has mean 0 across a batch.

## 3.4 VARIANCE ANALYSIS IN BACKWARD PASS

We consider backward propagation from the $L$th layer to the $(L-1)$th layer and focus on gradient propagation. Since, the gradient has just passed through the BN sub-layer of Lth layer, using (5) we get E($\Delta \mathbf{y}_L$) = 0. First, gradients go through the CONV sub-layer.

Under the following three assumptions: 1) elements in $\mathbf{W}_L^b$ are mutually independent and with the same distribution of mean 0, 2) elements in $\Delta \mathbf{y}_L$ are mutually independent and with the same distribution across a batch, and 3) $\Delta \mathbf{y}_L$ and $\mathbf{W}_L^b$ are independent of each other. Then, we get

$$Var(\Delta \hat{y}_{L-1,i}) = n_L' Var(\Delta y_{L,i}) Var(W_{L,i}^b), \text{ and } E(\Delta \hat{y}_{L-1,i}) = E(\Delta \mathbf{y}_{L,i}) = 0. \tag{11}$$

Next, gradients go through the ReLU sub-layer. It is assumed that the function applied to the gradient vector on passing through ReLU and the elements of gradient are independent of each other. Since the input in the forward pass was a shifted normal variate ($a = \frac{\beta_i}{\gamma_i}$), we get

$$E(\Delta \tilde{y}_{L-1,i}) = (0.5 + \frac{a}{\sqrt{2\pi}}) E(\Delta \hat{y}_{L-1,i}) = 0.0, \text{ and } Var(\Delta \tilde{y}_{L-1,i}) = (0.5 + \frac{a}{\sqrt{2\pi}}) Var(\Delta \hat{y}_{L-1,i}). \tag{12}$$

In the final step, gradients go through the BN sub-layer. If the standard normal variate, $\mathbf{z}$, to the BN sub-layer and the incoming gradients $\Delta \mathbf{y}$ are independent, we have $E(z_i \Delta y_{L-1,i}) = E(z_i) E(\Delta y_{L-1,i}) = 0$. The last equality holds since the mean of the standard normal variate is zero. The final result is

$$Var(\Delta y_{L-1,i}) = Var(\Delta y_{L,i}) \frac{n_L'}{n_{L-1}} \frac{Var(W_{L,i}^b)}{Var(W_{L-1,i}^f)} \frac{0.5 + \sqrt{\frac{1}{2\pi}} a}{0.5 + \sqrt{\frac{2}{\pi}} a + 0.5a^2 + \frac{a^3}{3\sqrt{2\pi}}}. \tag{13}$$

Note that the last product term in the derived formula is the term under consideration for checking gradient explosion or vanishing. The other two fractions are properties of the network, that compare two adjacent Layers. The skipped steps are given in Appendix B.

## 3.5 DISCUSSION

Initially, we set $\beta_i = 0$ and $\gamma_i = 1$ so that $a = 0$. Then, the last product term in the RHS of Eq. (13) is equal to one. Hence, if the weight initialization stays equal across all the layers, propagated gradients are maintained throughout the network. In other words, the BN simplifies the weight initialization job. For intermediate steps, we can estimate the gradient variance under simplifying assumptions that offer a simple minded view of gradient propagation. Note that, when $a = \frac{\beta}{\gamma}$ is small, the last product term is nearly equal to one. The major implication is that, the BN helps maintain gradients across the network, throughout the training, thus stabilizing optimization.

## 4 GRADIENT PROPAGATION THROUGH A RESNET BLOCK

### 4.1 RESNET BLOCK

The resnet blocks in the forward pass and in the gradient backpropagation pass are shown in Figs. 2 and 3, respectively. A residual network has multiple scales, each scale has a fixed number of residual blocks, and the convolutional layer in residual blocks at the same scale have the same number of filters. In the analysis, we adopt the model where the filter number increases $k$ times from one scale to the next one. Although no bottleneck blocks are explicitly considered here, our analysis holds for bottleneck blocks as well. As shown in Fig. 2, the input passes through a sequence of BN, ReLU and CONV sub-layers along the shortcut branch in the first residual block of a scale, which shapes the input to the required number of channels in the current scale. For all other residual blocks in the same scale, the input just passes through the shortcut branch. For all residual blocks, the input goes through the convolution branch which consists of two sequences of BN, ReLU and CONV sub-layers. We use a *layer* to denote a sequence of BN, ReLU and CONV sub-layers as used in the last section and $F$ to denote the compound function of one layer.

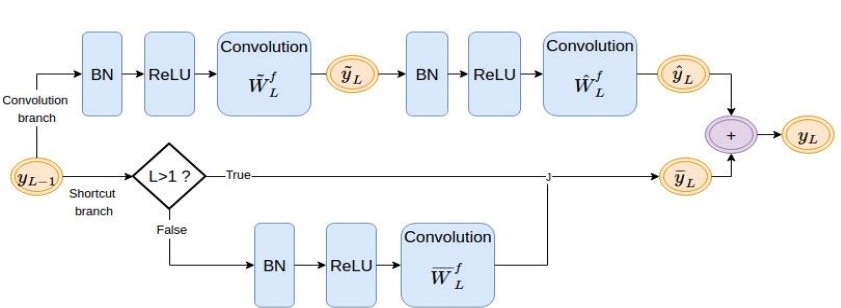

Figure 2: A residual block in the forward pass.

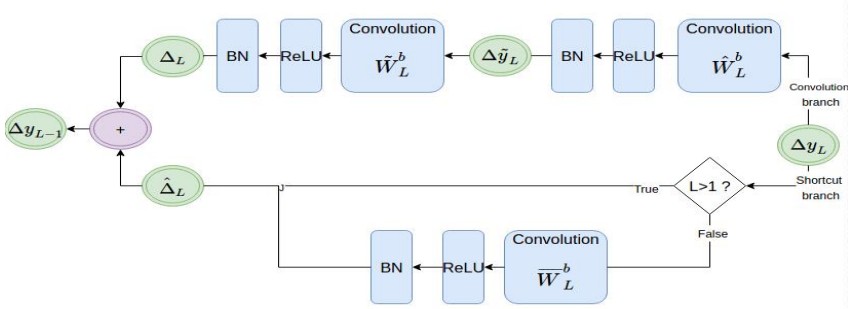

Figure 3: A residual block in the gradient backpropagation pass.

To simplify the computation of the mean and variance of $y_{L,i}$ and $\Delta y_{L,i}$, we assume that $a = \frac{\beta_i}{\gamma_i}$ is small ($<1$) across all the layers so that we can assume $a$ as constant for all the layers. We define the following two associated constants.

$$c_1 = 0.5 + \sqrt{\frac{1}{2\pi}}a \tag{14}$$

$$c_2 = 0.5 + \sqrt{\frac{2}{\pi}a + 0.5a^2 + \frac{a^3}{3\sqrt{2\pi}}} \tag{15}$$

which will be needed later.

### 4.2 VARIANCE ANALYSIS

As shown in Fig. 2, block $L$ is the $L$th residual block in a scale with its input $y_{L-1}$ and output $y_L$. The outputs of the first and the second BN-ReLU-CONV layers in the convolution branch are

$\tilde{y}_L = F(y_{L-1})$ and $\hat{y}_L = F(F(y_{L-1}))$, respectively. The weight vectors of the CONV sub-layer of the first and the second layers in the convolution branch of block $L$ are $\tilde{W}_L$ and $\hat{W}_L$, respectively. The weight vector in the shortcut branch of the first block is $\overline{W}_1$. The output of the shortcut branch is $\overline{y}_L$. For $L = 1$, we have $\overline{y}_1 = F(y_0)$, where $y_0$ is the output of last residual block of the previous scale. For L>1, we have $\overline{y}_L = y_{L-1}$. For the final output, we have

$$y_L = \overline{y}_L + \hat{y}_L. \tag{16}$$

For L>1, block $L$ receives an input of size $n_s$ in the forward pass and an input gradient of size $n'_s$ in the backpropagation pass. Since block 1 receives its input $y_0$ from the previous scale, it receives an input of size $\frac{n_s}{k}$ in the forward pass.

By assuming $\overline{y}_L$ and $\hat{y}_L$ are independent, we have

$$Var(y_{L,i}) = Var(\overline{y}_{L,i}) + Var(\hat{y}_{L,i}). \tag{17}$$

We will show how to compute the variance of $y_{L,i}$ step by step in Appendix C for $L = 1, \cdots, N$. When $L = N$, we obtain

$$Var(y_{N,i}) = c_2 n_s (\sum_{J=2}^{N} Var(\hat{W}_{J,i}^f) + \frac{1}{k}(Var(\overline{W}_{1,i}^f) + Var(\hat{W}_{1,i}^f))), \tag{18}$$

where $c_2$ is defined in Eq. (15).

We use $\Delta$ as prefix in front of vector representations at the corresponding positions in forward pass to denote the gradient in Fig. 3 in the backward gradient propagation. Also, as shown in Fig. 3, we represent the gradient vector at the tip of the convolution branch and shortcut branch by $\Delta_L$ and $\hat{\Delta}_L$ respectively. As shown in the figure, we have

$$\Delta y_{L-1} = \hat{\Delta}_L + \Delta_L \tag{19}$$

A step-by-step procedure in computing the variance of $\Delta y_{L-1,i}$ is given in Appendix D. Here, we show the final result below:

$$Var(\Delta y_{L-1,i}) = \left(1 + (\frac{c_1}{c_2})^2 \frac{Var(\tilde{W}_{L,i}^b)}{Var(\tilde{W}_{L,i}^f)} \frac{Var(\hat{W}_{L,i}^b)}{\sum_{J=2}^{L-1} Var(\hat{W}_{J,i}^f) + \frac{1}{k}(Var(\overline{W}_{1,i}^f) + Var(\hat{W}_{1,i}^f))}\right) Var(\Delta y_{L,i}). \tag{20}$$

### 4.3 DISCUSSION

We can draw two major conclusions from the analysis conducted above. First, it is proper to relate the above variance analysis to the gradient vanishing and explosion problem. The gradients go through a BN sub-layer in one residual block before moving to the next residual block. As proved in Sec. 3, the gradient mean is zero when it goes through a BN sub-layer and it still stays at zero after passing through a residual block. Thus, if it is normally distributed, the probability of the gradient values between $\pm$ 3 standard deviations is 99.7%. A smaller variance would mean lower gradient values. In contrast, a higher variance implies a higher likelihood of discriminatory gradients. Thus, we take the gradient variance across a batch as a measure for stability of gradient backpropagation.

Second, recall that the number of filters in each convolution layer of a scale increases by $k$ times with respect to its previous scale. Typically, $k = 1$ or 2. Without loss of generality, we can assume the following: the variance of weights is about equal across layers, $c_1/c_2 \approx 1$, and $k = 2$. Then, Eq. (20) can be simplified to

$$Var(\Delta y_{L-1,i}) \approx \frac{L}{L-1} Var(\Delta y_{L,i}). \tag{21}$$

We see from above that the change in the gradient variance from one residual block to its next is little. This is especially true when the $L$ value is high. This point will be further discussed in the next section.

## 4.4 EXPERIMENTAL VERIFICATION

We trained a Resnet-15 model that consists of 15 residual blocks and 3 scales on the CIFAR-10 dataset, and checked the gradient variance across the network throughout the training. We plot the mean of the gradient variance and the $l_2$-norm of the gradient at various residual block locations in Figs. 4 and 5, respectively, where the gradient variance is calculated for each feature across one batch. Since gradients backpropagate from the output layer to the input layer, we should read each plot from right to left to see the backpropagation effect. The behavior is consistent with our analysis. There is a gradual increase of the slope across a scale. The increase in the gradient variance between two residual blocks across a scale is inversely proportional to the distance between the residual blocks and the first residual block in the scale. Also, there is a dip in the gradient variance value when we move from one scale to its previous scale. Since the BN sub-layer is used in the shortcut branch of the first block of a scale, it ensures the decrease of the gradient variance as it goes from one scale to another. Some other experiments that we conducted to support our theory can be found in Appendix E.

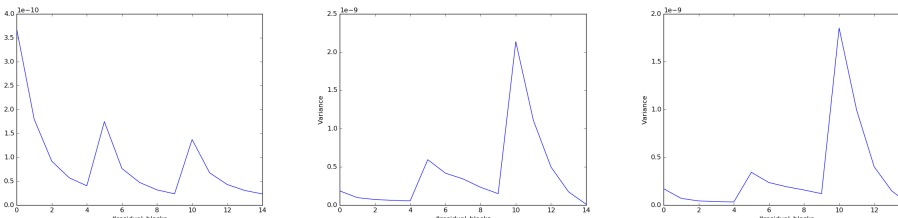

Figure 4: The mean of the gradient variance as a function of the residual block position at Epoch 1 (left), Epoch 25000 (middle) and Epoch 50000 (right).

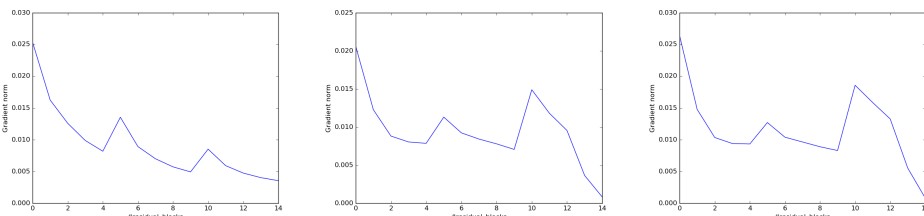

Figure 5: The $l_2$ norm of the gradient as a function of the residual block position at Epoch 1 (left), Epoch 25000 (middle) and Epoch 50000 (right).

## 5 WIDTH VERSUS DEPTH IN RESNETS

Veit et al. (2016) showed that the paths which gradients take through a ResNet are typically far shorter than the total depth of that network. For this reason, they introduced the "effective depth" idea as a measure for the true length of these paths. They showed that almost all of gradient updates in the training come from paths of 5-17 modules in their length. Wu et al. (2016) also presented a similar concept. That is, residual networks are actually an ensemble of various sub-networks and it echoes the concept of effective depth. Overall, the main point is that some residual blocks stay dormant during gradient backpropagation.

Based on our analysis in Sec. 4, the gradient variance should increase by $L/(L-1)$ after passing through a residual block, where $(L-1)$ is the distance of the current residual block from the first residual block in a scale. Thus, the gradient variance should not change much as the gradient backpropagates through the chain of residual networks at the far end of a scale if we use a residual network of high depth. Since a lower gradient variance value implies non-varying gradient values, it supports the effective path concept as well. As a result, the weights in the residual blocks see similar gradient variation without learning much discriminatory features. In contrast, for networks of lower depth , the gradient variance changes more sharply as we go from one residual block to another. As

a result, all weights present in the network are used in a more discriminatory way, thus leading to better learning.

We compare the performance of the following three Resnet models:

1. Resnet-99 with 99 resnet blocks,
2. Resnet-33 with 33 resnet blocks and tripled filter numbers in each resnet block,
3. Resnet-9 with 9 resnet blocks and 11 times filter numbers in each resnet block.

Note that the total filter numbers in the three models are the same for fair comparison. We trained them on the CIFAR-10 dataset.

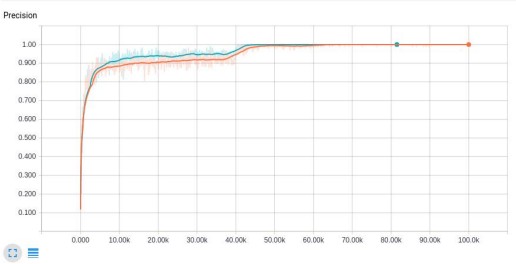

Figure 6: Comparison of training accuracy for Resnet-99 (red) and Resnet-9 (blue).

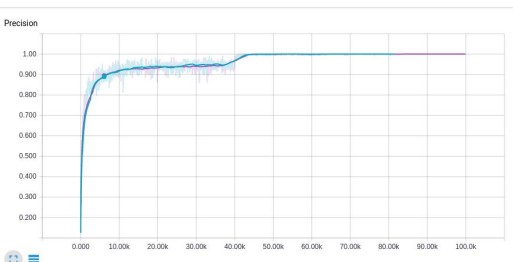

Figure 7: Comparison of training accuracy for Resnet-33 (violet) and Resnet-9 (blue).

First, we compare the training accuracy between Resent-9 and Resnet-99 in Fig. 6 and that between Resent-9 and Resnet-33 in Fig. 7, where the horizontal axis shows the epoch number. We see that Resnet-9 reaches the higher accuracy faster than both Resnet-99 and Resnet-33, yet the gap between Resnet-9 and Resnet-33 is smaller than that between Resnet-9 and Resnet-99. This supports our claim that a shallow-wide Resnet learns faster than a deep-narrow Resnet. Next, we compare their test set accuracy in Table 1. We see that Resnet-9 has the best performance while Resnet-99 the worst. This is in alignment with our prediction and the experimental results given above.

| Model | Final accuracy |
|---|---|
| Resnet with 99 resnet blocks | 93.4% |
| Resnet with 33 resnet blocks | 93.8% |
| Resnet with 9 resnet blocks | 94.4% |

Table 1: Comparison of test accuracy of three Resnet models at epoch 100,000

Furthermore, we plot the mean of the gradient variance, calculated for each feature across one batch, as a function of the residual block index at epochs 1, 25,000 and 50,000 in Figs. 8, 9 and 10, respectively, where the performance of Resnet-99, Resnet-33 and Resnet-9 is compared. We observe that the gradient variance does not change much across a batch as it passes through the residual blocks present at the far end of a scale in Resnet-99. For Resnet-33, there are fewer resnet blocks at the far end of a scale that stay dormant. We can also see clearly the gradient variance changes more

sharply during gradient backpropagation in resnet-9. Hence, the residual blocks present at the end of a scale have a slowly varying gradient passing through them in Resnet-99, compared to Resnet-33 and Resnet-9. These figures show stronger learning performance of shallower but wider resnets.

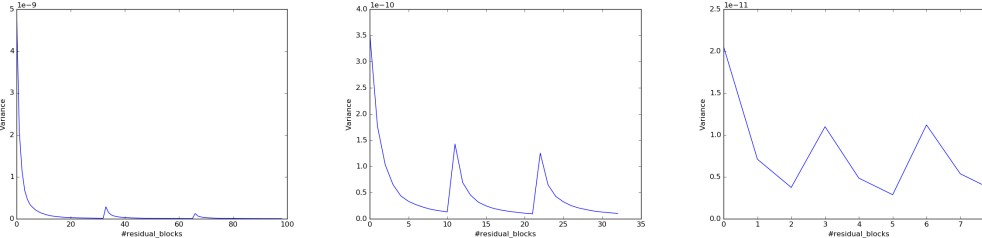

Figure 8: The gradient variance as a function of the residual block index during backpropagation in Resnet-99 (left), Resnet-33 (middle) and Resnet-9 (right) at Epoch 1.

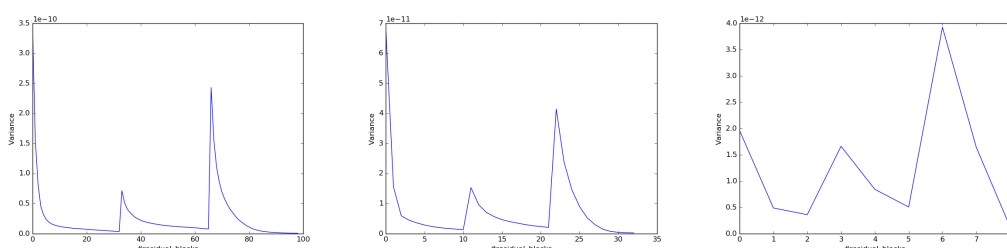

Figure 9: The gradient variance as a function of the residual block index during backpropagation in Resnet-99 (left), Resnet-33 (middle) and Resnet-9 (right) at Epoch 25000.

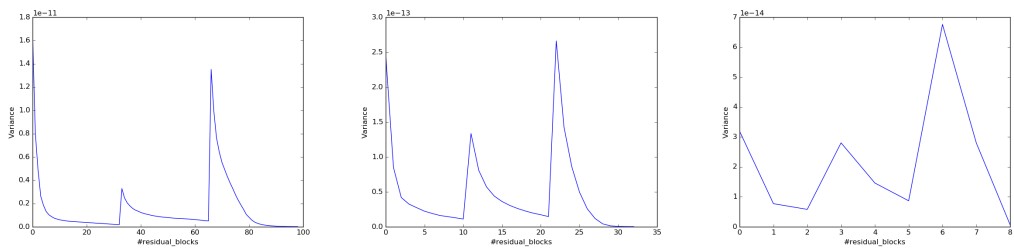

Figure 10: The gradient variance as a function of the residual block index during backpropagation in Resnet-99 (left), Resnet-33 (middle) and Resnet-9 (right) at Epoch 50000.

## 6   CONCLUSION AND FUTURE WORK

Batch normalization (BN) is critical to the training of deep residual networks. Mathematical analysis was conducted to analyze the BN effect on gradient propagation in residual network training in this work. We explained how BN and residual branches work together to maintain gradient stability across residual blocks in back propagation. As a result, the gradient does not explode or vanish in backpropagation throughout the whole training process. Furthermore, we applied this mathematical analysis to the decision on the residual network architecture - whether it should be deeper or wider. We showed that a slowly varying gradient across residual blocks results in lower learning capability and deep resnets tend to learn less than their corresponding wider form. The wider resnets tend to use their parameter space better than the deeper resnets.

The Saak transform has been recently introduced by Kuo & Chen (2017), which provides a brand new angle to examine deep learning. The most unique characteristics of the Saak transform approach

is that neither data labels nor backpropagation is needed in training the filter weights. It is interesting to study the relationship between multi-stage Saak transforms and residual networks and compare their performance in the near future.

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

## APPENDIX A

We apply the ReLU to the output of a BN layer, and show the step-by-step procedure in calculating the variance and the mean of the output of the ReLU operation. In the following derivation, we drop the layer and the element subscripts (i.e., $L$ and $i$) since there is no confusion. It is assumed that scaling factors, $\beta$ and $\gamma$, in the BN are related such that $a = \beta/\gamma$ is a small number and the standard normal variable $z$ has a nearly uniform distribution in $(-a, 0)$. Then, we can write the shifted Gaussian variate due to the BN operation as

$$\gamma z + \beta = \gamma(z + a). \tag{22}$$

Let $y = ReLU(z + a)$. Let a > 0. We can write

$$E(y) = P(z < -a)E(y|z < -a) + P(-a < z < 0)E(y|-a < z < 0) + P(z > 0)E(y|z > 0). \tag{23}$$

The first right-hand-side (RHS) term of Eq. (23) is zero since $y = 0$ if $z < -a$ due to the ReLU operation. Thus, $E(y|z < -a) = 0$. For the second RHS term, $z$ is uniformly distributed with probability density function equal to $a^{-1}$ in range (-a, 0) if $0 < a << 1$. Then, we have

$$P(-a < z < 0) = \frac{a}{\sqrt{2\pi}}, \quad \text{and} \quad E(y|-a < z < 0) = \frac{a}{2}. \tag{24}$$

For the third RHS term, $P(z > 0) = 0.5$. Besides, $z > 0$ is half-normal distributed. Thus, we have

$$E(y|z > 0) = E(|z|) + a = \sqrt{\frac{2}{\pi}} + a. \tag{25}$$

Based on the above results, we get

$$E(y) = \frac{1}{\sqrt{2\pi}} + \frac{a}{2} + \frac{a^2}{2\sqrt{2\pi}}. \tag{26}$$

Similarly, we can derive a formula for $E(y^2)$ as

$$\begin{aligned} E(y^2) &= P(z < -a)E(y^2|z < -a) + P(-a < z < 0)E(y^2|-a < z < 0) \\ &\quad + P(z > 0)E(y^2|0 < z < a). \end{aligned} \tag{27}$$

For the first RHS term of Eq. (27), we have $E(y^2|z < -a) = 0$ due to the ReLU operation. For the second RHS term of Eq. (27), $z$ is uniformly distributed with probability density function $a^{-1}$ for -a<z<0 so that $P(-a < z < 0) = \frac{a}{\sqrt{2\pi}}$ and $E(y^2|-a < z < 0) = \frac{a^2}{3}$. For the third RHS term $P(z > 0) = 0.5$ for $z > 0$. The random variable $z > 0$ is half normal distributed so that

$$E(y^2|z > 0) = E((|z| + a)^2) = E(|z|^2) + a^2 + 2aE(|z|) = a^2 + 2\sqrt{\frac{2}{\pi}}a + 1. \tag{28}$$

Then, we obtain

$$E(y^2) = 0.5 + \sqrt{\frac{2}{\pi}}a + 0.5a^2 + \frac{a^3}{3\sqrt{2\pi}} \tag{29}$$

We can follow the same procedure for $a < 0$. The final results are summarized below.

$$E(ReLU(\gamma z + \beta)) = \gamma E(y) \quad , \text{and} \quad E((ReLU(\gamma z + \beta))^2) = \gamma^2 E(y^2), \tag{30}$$

where $E(y)$ and $E(y^2)$ are given in Eqs. (26) and (29), respectively.

## APPENDIX B

- We assumed that the function(F) applied by ReLU to the gradient vector and the gradient elements are independent of each other. Function F is defined as

$$F(\Delta y) = \Delta y I_{y>0}$$

where $\Delta y$ denotes input gradient in gradient backpropagation and y denotes the input activation during forward pass to the ReLU layer. Coming back to our analysis, since $\tilde{y}_{L-1,i}$

is a normal variate shifted by $a$, the probability that the input in forward pass to the ReLU layer, i.e. $\tilde{y}_{L-1,i}$ is greater than 0 is

$$P(\tilde{y}_{L-1,i} > 0) = 0.5 + \frac{a}{\sqrt{2\pi}}$$

Thus, $E(F(\Delta\hat{y}_{L-1,i})) = E(\Delta\hat{y}_{L-1,i})\, P(\tilde{y}_{L-1,i} > 0)$, and so

$$E(\Delta\tilde{y}_{L-1,i}) = (0.5 + \frac{a}{\sqrt{2\pi}})\, E(\Delta\hat{y}_{L-1,i})$$

Similarly, we can solve for $Var(\Delta\tilde{y}_{L-1,i})$ and thus, get Eq. (12).

- First, using eq 5 and the assumption that the input standard normal variate in forward pass and the input gradient in gradient pass are independent, we have

$$Var(\Delta y_{L-1,i}) = \frac{\gamma_i^2}{Var(y_{L-1,i})} Var(\Delta\tilde{y}_{L-1,i}) \tag{31}$$

$$= \frac{\gamma_i^2}{Var(y_{L-1,i})} n_l'(0.5 + \frac{a}{\sqrt{2\pi}}) Var(W_{L,i}^b) Var(\Delta y_{L,i}). \tag{32}$$

Then, using Eq. (10) for $Y_{L-1}$ (yet with $L$ replaced with $L-1$), we can get Eq. (13).

## APPENDIX C

For $L = 1$, $\overline{y}_1 = F(y_0)$. Since the receptive field for the last scale is $k$ times smaller, we get the following from Eq. (10),

$$Var(\overline{y}_{1,i}) = c_2 \frac{n_s}{k} Var(\overline{W}_{1,i}^f). \tag{33}$$

Also, since $\hat{y}_1 = F(F(y_0))$, we have

$$Var(\hat{y}_{1,i}) = c_2 \frac{n_s}{k} Var(\hat{W}_{1,i}^f)$$

based on Eq. (10). Therefore, we get

$$Var(y_{1,i}) = c_2 \frac{n_s}{k} Var(\overline{W}_{1,i}^f) + c_2 \frac{n_s}{k} Var(\hat{W}_{1,i}^f) \tag{34}$$

$$= c_2 \frac{n_s}{k} (Var(\overline{W}_{1,i}^f) + Var(\hat{W}_{1,i}^f)). \tag{35}$$

For $L = N > 1$, the input just passes through the shortcut branch. Then,

$$Var(\overline{y}_{N,i}) = Var(y_{N-1,i})$$

Also, since $\hat{y}_N = F(F(y_{N-1}))$, we have

$$Var(\hat{y}_{N,i}) = c_2\, n_s\, Var(\hat{W}_{N,i}^f).$$

due to using Eq. (10). Thus,

$$Var(y_{N,i}) = Var(y_{N-1,i}) + c_2\, n_s\, Var(\hat{W}_{N,i}^f). \tag{36}$$

Doing this recursively from $L = 1$ to $N$, we get

$$Var(y_{N,i}) = c_2 n_s \left( \sum_{J=2}^{N} Var(\hat{W}_{J,i}^f) + \frac{1}{k}(Var(\overline{W}_{1,i}^f) + Var(\hat{W}_{1,i}^f)) \right). \tag{37}$$

## APPENDIX D

For block $L = N > 1$, the gradient has to pass through two BN-ReLU-Conv Layers in convolution branch. Since, the receptive field doesn't change in between the two BN-ReLU-Conv Layers in the convolution branch of the block, we use Eq. (13) and find that for same receptive field between the two layers i.e. $n_L = n'_{L-1}$ ,

$$Var(\Delta \tilde{y}_{L,i}) = \frac{c_1}{c_2} \frac{Var(\hat{W}^b_{L,i})}{Var(\tilde{W}^f_{L,i})} Var(\Delta y_{L,i}). \tag{38}$$

When gradient passes through the first BN-ReLU-Conv Layer, the variance of the forward activation that BN component sees is actually the variance of the output of previous block. Hence, using $Var(y_{L-1,i})$, which is the output of previous residual block, in place of the denominator in Eq. (31), we get

$$Var(\Delta_{L,i}) = \frac{c_1}{c_2} \frac{Var(\tilde{W}^b_{L,i})}{\sum_{J=2}^{L-1} Var(\hat{W}^f_{J,i}) + \frac{1}{k}(Var(\overline{W}^f_{1,i}) + Var(\hat{W}^f_{1,i}))} Var(\Delta \tilde{y}_{L,i}), \tag{39}$$

We assume that $\hat{\Delta}_L$ and $\Delta_L$ are independent of each other. Since we are calculating for Block L>1 where there is no BN-ReLU-Conv Layer in shortcut branch, we have $Var(\hat{\Delta}_{L,i}) = Var(\Delta y_{L,i})$. As,
$$Var(\Delta y_{L-1,i}) = Var(\Delta_{L,i}) + Var(\hat{\Delta}_{L,i}).$$
Finally, we obtain

$$Var(\Delta y_{L-1,i}) = \left(1 + (\frac{c_1}{c_2})^2 \frac{Var(\tilde{W}^b_{L,i})}{Var(\tilde{W}^f_{L,i})} \frac{Var(\hat{W}^b_{L,i})}{\sum_{J=2}^{L-1} Var(\hat{W}^f_{J,i}) + \frac{1}{k}(Var(\overline{W}^f_{1,i}) + Var(\hat{W}^f_{1,i}))}\right) Var(\Delta y_{L,i}). \tag{40}$$

## APPENDIX E

We compared two variations of residual network with the original model. The models were trained on CIFAR-10. The models compared were

- $Model_1$: Residual network with BN and residual branches
- $Model_2$: Residual network with BN but residual branches removed
- $Model_3$: Residual network with residual branches but BN removed

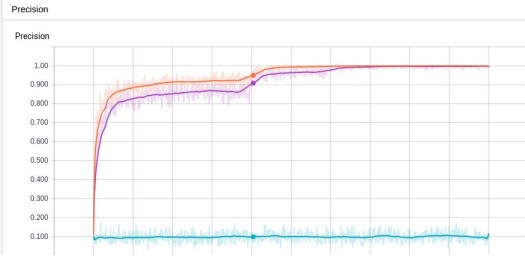

Figure 11: Comparison of training accuracy for $Model_1$(red), $Model_2$(violet) and $Model_3$ (blue).

All the models had 15 residual blocks, 5 in each scale. The parameters of each model were initialized similarly and were trained for same number of epochs. The weights were initialized with xavier initialization and the biases were initialized to 0. First, we compare the training accuracy among the three models in Fig. 11, where the horizontal axis shows the epoch number. We see that $Model_1$ reaches higher accuracy faster than the other two models. However, $Model_2$ isn't far behind. But

| Model | Final accuracy |
|-------|----------------|
| $Model_1$ | 92.5% |
| $Model_2$ | 90.6% |
| $Model_3$ | 9.09% |

Table 2: Comparison of test accuracy of three Resnet models.

$Model_3$, which has BN removed, doesn't learn anything. Next, we compare their test set accuracy in Table 2. We see that $Model_1$ has the best performance while $Model_2$ isn't far behind.

Furthermore, we plot the mean of the gradient variance, calculated for each feature across one batch, as a function of the residual block index at epochs 25,000, 50,000 and 75,000 in Figs. 12, 13 and 14, respectively, where the performance of $Model_1$ and $Model_2$ is compared. We observe that the gradient variance also stays within a certain range, without exploding or vanishing, in case of $Model_2$. However, the change in gradient variance across a scale doesn't follow a fixed pattern compared to $Model_1$. We also plot a similar kind of plot for $Model_3$ at epoch-1 in Fig 15. We observed gradient explosion, right from the start, in case of $Model_3$ and the loss function had quickly become undefined. This was the reason, why $Model_3$ didn't learn much during the course of training.

This experiment shows that BN plays a major role in stabilizing training of residual networks. Even though we remove the residual branches, the network still tries to learn from the training set, with its gradient fixed in a range across layers. However, removing BN hampers the training process right from the start. Thus, we can see that batch normalization helps to stop gradient vanishing and explosion throughout training, thus stabilizing optimization.

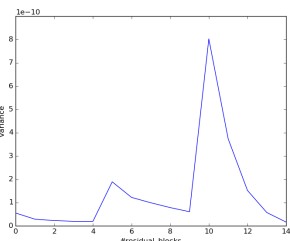 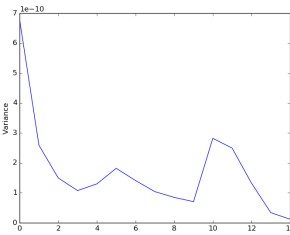

Figure 12: The gradient variance as a function of the residual block index during backpropagation in $Model_1$ (left), and $Model_2$ (right) at Epoch 25000.

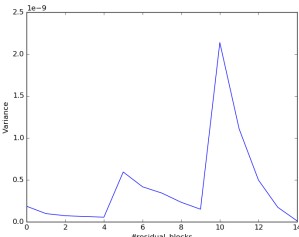 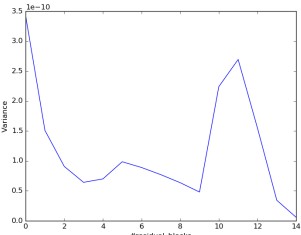

Figure 13: The gradient variance as a function of the residual block index during backpropagation in $Model_1$ (left), and $Model_2$ (right) at Epoch 50000.

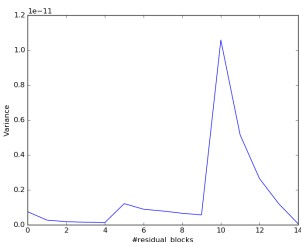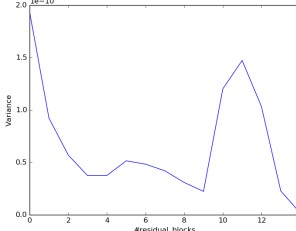

Figure 14: The gradient variance as a function of the residual block index during backpropagation in $Model_1$ (left), and $Model_2$ (right) at Epoch 75000.

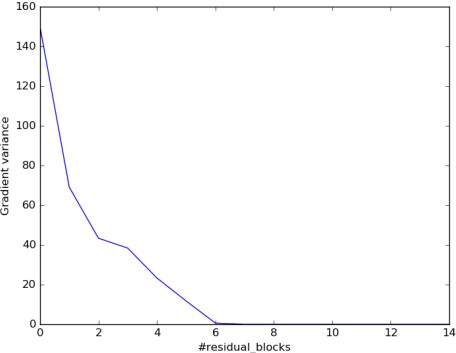

Figure 15: Gradient explosion observed during back propagation in $Model_3$ at epoch-1