# OpenReview forum: "ANALYSIS ON GRADIENT PROPAGATION IN BATCH NORMALIZED RESIDUAL NETWORKS"
_ICLR.cc/2018/Conference — Reject_

### Official Review · AnonReviewer2 · 2017-11-08
**Fatal Mathematical flaw and other big problems**

**Rating:** 1
**Confidence:** 5

**Review:**

This paper attempts to analyze the gradient flow through a batchNorm-ReLU ResNet and make suggestions for reducing gradient explosion.

Firstly, the paper has a fatal mathematical flaw. Consider equation (10). There, you show the variance of y_{L,i} taken over BOTH random weights AND the batch. Now consider equation (32). In that equation, Var(y_{L,i}) appears in the denominator but this variance is taken over ONLY the batch and NOT the random weights. This Var(y_{L,i}) came from batch normalization, which divides its incoming activation values by their standard deviation. However, batch normalization only sees the variation in the activations given to it by a SPECIFIC set of weights. It does not know about the random variation of the weights because that randomness is in a sense a superstructure imposed on the network that the network operations themselves cannot see. Therefore, your substitution and therefore equation (13) is incorrect. If you replace the variance in equation (32) by the correct value, you will get a very different result from which very different (and very interesting!) conclusions can be drawn.

Secondly, in section 4, your analysis depends on the specific type of ResNet you chose. Specifically, when transitioning from one "scale" to the next, you chose to insert not just a convolutional layer, but also a batch normalization and ReLU layer on the residual path. To achieve scale transitions, in general, people use a single convolutional layer with 1*1 receptive field on the residual path. It is not a problem in itself to use a nonstandard architecture, but you do not discuss how your results would generalize to other ResNet architectures. Therefore your results have very limited relevance. (Note that again, of course, your results are corrupted by the variance problem I described earlier.)

Finally, with regards to section 5, let me be honest. (I hope that my area chair agrees with me that honesty is the best and kindest policy.) This section makes no sense. You do not understand the work by Veit et al. You do not know how to interpret gradient variances. While I won't be able to dicuss "gradient variance" as a concept in full detail in this review, here's a quick summary. (A) Veit et al. argued that a deep ResNet behaves as an ensemble of shallower networks as long as the gradient flowing through the residual paths is not larger than the gradient flowing through the skip paths. (B) The exploding gradient problem refers to the size of the gradient growing exponentially. The vanishing gradient problem refers to the size of the gradient shrinking exponentially. This can make it difficult to train the network. See "DEEP INFORMATION PROPAGATION" by Schoenholz et al. from ICLR 2017 to learn more about how gradient explosion can arise. (C) For a neural network to model a ground truth function exactly, the gradients of the network with respect to the input data have to match the gradients of the ground truth function with respect to the input. From observations (A) through (C), we can derive three guidelines for gradient conditioning: (A) have the gradient flowing through residual paths be not too small relative to the gradient flowing through skip paths, (B) have the gradient not grow or shrink exponentially with too large a rate, (C) have the data gradient match that of the ground truth function. However, you seem to be arguing that it is a problem if the gradient scale does increases too little from one residual block to the next. I am not aware of an established argument that this is indeed a problem. To be fair, one might make an argument as follows: "the point of deep nets is to be expressive, expressiveness of a layer relates to the spetrum of the layer-Jacobian, a small increase in gradient scale implies the layer-Jacobian has many similar singular values, therefore a small increase in gradient scale implies low expressiveness of the layer, therefore the layer is pathological". However, much more analysis, detail and care would be required to make this argument successfully. In any case, I also don't think that was the argument you were trying to make. Note that after I skimmed through the submissions to this conference, there seem to be interesting papers on the topic of gradients. Those papers plus the references provided in those papers should provide a good introduction to the topic of gradients in neural networks.

Other comments:
 - your notation is quite sloppy and may have lead to errors. Example: in the beginning of section 4, you say that from one "scale" to the next, the filter number increases k times. But in appendix C you say "Since the receptive field for the last scale is k times smaller". So is k the change in the receptive field size or the filter number of both? I would strongly recommend using dedicated variables to denote the width of the receptive field in each convolutional layer, the height of the receptive field in each convolutional layer as well as the filter number and then express all assumptions made in equation form.
- Equation (20) deals with the change of gradient variance within a scale. Where is the equation that shows the change of gradient variance between scales?
- I would suggest making all derivations in appendices A through D much more detailed.

---

### Official Review · AnonReviewer3 · 2017-11-28
**This paper analyzed the effect of batch normalization (BN) on gradient backpropagation in residual networks (ResNets). However, the results are not well justified.**

**Rating:** 4
**Confidence:** 5

**Review:**

Summary:
This paper analyzed the effect of batch normalization (BN) on gradient backpropagation in residual networks (ResNets). The authors demonstrate that BN can confine the gradient variance to a certain range in each residual block. The analysis is extended to discuss the trade-off between the depth and width of residual networks. However, the effect of BN in ResNets is still not clear and some definitions in this paper are confusing.

Strengths:
1. This paper conducted mathematical analysis on the effect of batch normalization (BN) in residual networks during back-propagation.

2. The authors demonstrated that BN confined the gradient variance to a certain range in each residual block.

3. The authors discussed the tradeoff between the depth and width of residual networks based on the analysis of BN.

Weak points:
1. The motivation of the analysis on the effect of BN in residual network is not clear. Compared to the plain network with BN, the gradient vanishing/explosion problem has been largely addressed by the shortcut of identity mapping. After reading the paper, it is still not clear what kind of role the BN plays for addressing this issue, especially when compared to the effect of identity mapping.

2. There seems a definition error in the first paragraph of Section 3.1. Should $\delta x$ be the batch of input gradient and $\delta \tilde x$ be the batch of output gradient?

3. In Section 3.1, what does “the standard normal variate of x is z” mean?

4. The definition of x_i in Eqn. (4) is very confusing, which makes the paper hard to follow. Here, the subscript x_i should be the i-th channel of input feature map rather than the i-th example in a mini-batch. However, in the original BN paper, all the gradients are computed w.r.t. the i-th example in a mini-batch. So, how to transform the formulas in the original BN paper to the gradient w.r.t. a specific channel like Eqn. (4). More details should be provided.

5. In Section 3.2, it is strange that the authors consider a basic block containing  BN and ReLU, followed by a convolution layer. However, in general deep learning setting,  the BN and ReLU is often put after a convolution layer. Please explain more on this point.

6. In Section 3.3, the authors assume that the weights of convolution layer have zero-mean, because the mean of input/output gradient is zero. However, it does not mean that the gradient w.r.t. the weights has zero-mean and the gradient will introduce a distribution bias in the weights.

7. In Section 5, the authors argued that the sharper gradient variance changes resulted in more discriminatory learning. However, this is not well justified.

---

### Official Review · AnonReviewer1 · 2017-11-30
**Interesting analysis of gradient propagation from Resnets+Batch Normalization**

**Rating:** 4
**Confidence:** 4

**Review:**

This manuscript is fairly well-written, and discusses how the batch normalization step helps to stabilize the scale of the gradients.  Intriguingly, the analysis suggests that using a shallower but wider resnet should provide competitive performance, which is supported by empirical evidence.  This work should help elucidate the structure in the learning, and help to support efforts to improve both learning algorithms and the architecture.

Pros:
Clean, simple analysis
Empirical support suggests that theory captures reasonable effects behind learning

Cons:
The reasonableness of the assumptions used in the analysis needs a more careful analysis.  In particular, the assumption that all weights are independent is valid only at the first random iteration.  Therefore, the utility of this theory during initialization seems reasonable, but during learning the theory seems quite tenuous.  I would encourage the authors to discuss their assumptions, and talk about how the math would change as a result of relaxing the assumptions.
The empirical support does provide evidence that the theory is reasonable.  However, it is limited to a single dataset.  It would be nice to see that the effect happens more generally. Second, it is clear that shallow+wide networks may be better than deep+narrow networks, but it's not clear about how the width is evaluated and supported. I would encourage the authors to do more extensive experiments and evaluate the architecture further.

Revision:
Upon examining the comments of the other reviews, I have agreed with several of their points and it is necessary to increase the explanation of the mathematical points.  I encourage the authors to address these comments and revise their work.

---

### Decision · Program_Chairs · 2018-01-29
**ICLR 2018 Conference Acceptance Decision**

**Decision:**

Reject

**Comment:**

Two of the reviewers liked the intent of the paper -- to analyze gradient flow in residual networks and understand the tradeoffs between width and depth in such networks.  However, all reviewers flagged a number of problems in the paper, and the authors did not participate in the discussion period.

Pros:
+ Interesting analysis suggests wider, shallower ResNets should outperform narrower, deeper ResNets, and empirical results support the analysis.

Cons:
- Independence assumption on weights is not valid after any weight updates.
- The notation is not as clear as it should be.
- Empirical results would be more convincing if obtained on several tasks.
- The architecture analyzed in the paper is not standard, so it isn't clear how relevant it is for other practitioners.
- Analysis and paper should take into account other work in this area, e.g. Veit et al., 2016 and Schoenholz et al., 2017.